# Entanglement Renyi Entropy of Two Disjoint Intervals for Large *c* Liouville Field Theory

**DOI:** 10.3390/e24121758

**Published:** 2022-11-30

**Authors:** Jun Tsujimura, Yasusada Nambu

**Affiliations:** Department of Physics, Nagoya University, Chikusa, Nagoya 464-8602, Japan

**Keywords:** semi-classical limit, Liouville field theory

## Abstract

Entanglement entropy (EE) is a quantitative measure of the effective degrees of freedom and the correlation between the sub-systems of a physical system. Using the replica trick, we can obtain the EE by evaluating the entanglement Renyi entropy (ERE). The ERE is a *q*-analogue of the EE and expressed by the *q* replicated partition function. In the semi-classical approximation, it is apparently easy to calculate the EE because the classical action represents the partition function by the saddle point approximation and we do not need to perform the path integral for the evaluation of the partition function. In previous studies, it has been assumed that only the minimal-valued saddle point contributes to the EE. In this paper, we propose that all the saddle points contribute comparably but not necessarily equally to the EE by dealing carefully with the semi-classical limit and then the q→1 limit. For example, we numerically evaluate the ERE of two disjoint intervals for the large *c* Liouville field theory with q∼1. We exploit the BPZ equation with the four twist operators, whose solution is given by the Heun function. We determine the ERE by tuning the behavior of the Heun function such that it becomes consistent with the geometry of the replica manifold. We find the same two saddle points as previous studies for q∼1 in the above system. Then, we provide the ERE for the large but finite *c* and the q∼1 in case that all the saddle points contribute comparably to the ERE. In particular, the ERE is the summation of these two saddle points by the same weight, due to the symmetry of the system. Based on this work, it shall be of interest to reconsider EE in other semi-classical physical systems with multiple saddle points.

## 1. Introduction

Evaluating the effective degrees of freedom of a physical system is a fundamental problem in physics. It is helpful to determine the phases of quantum many-body systems or to study the holographic principle, which states that the degrees of freedom of a gravitational system are equal to those of a system that is one dimension lower compared to the gravitational system. Entanglement entropy (EE) is a quantitative measure of the effective degrees of freedom and the correlation between the sub-systems of a physical system; thus, it has been investigated from viewpoints of thermodynamics, statistical mechanics, and information theory. Generally, the difficulty in estimating the value of EE depends on the complexity of the structure of a theory or the form of the sub-systems. Therefore, despite, the difficulty in evaluating the EE of general quantum field theory, EEs of two-dimensional conformal field theories are well studied owing to their abundant symmetries. Particularly, the global conformal symmetry determines the EE of a single interval regardless of the intricacies of the theories. However, when we deal with a two disjoint intervals sub-system, it is hard to evaluate the EE unless it is a simple theory such as the free field [1].

The entanglement Renyi entropy (ERE) is a *q*-analogue of the EE. The ERE $S_{A}\left(q\right)$ of the sub-system *A* is defined as
(1)$$\begin{array}{c} S_{A}\left(q\right)=\frac{1}{1-q}\log{\mathrm{tr}}_{A}\left(\rho_{A}^{\;q}\right), \end{array}$$
where $\rho_{A}$ is the partial density matrix on *A*. The partial density matrix is normalized as ${\mathrm{tr}}_{A}\left(\rho_{A}\right)=1$, and then the EE can be defined as $S_{A}=\lim_{q\to 1}S_{A}\left(q\right)$. The ERE is rewritten as
(2)$$\begin{array}{c} S_{A}\left(q\right)=\frac{1}{1-q}\left(\log Z_{A}\left(q\right)-q\log Z\right), \end{array}$$
where $Z_{A}\left(q\right)$ and *Z* denote the partition function of the *q*-replicated theory and that of the original theory, respectively [2]. The Liouville conformal field theory (CFT) has preferable properties for this formulation, which is studied in the context of the non-critical string theory, higher dimensional theory, etc. [3]. The Liouville CFT exhibits the semi-classical limit as the large *c* limit. In this limit, the evaluation of EREs is easier because the saddle points of the path integral represent the respective partition functions. Previous studies have reported that there exist two saddle points for $Z_{A}\left(q\right)$ for the two disjoint intervals system in the case of the large *c* Liouville CFT with q∼1, or in the adjacent interval limit [4,5,6]. Then, it has been assumed that only the minimal valued saddle point contributes to $Z_{A}\left(q\right)$.

In this paper, we numerically calculate the ERE for q∼1 using the Heun function. The Liouville CFT has postulated that the correlation functions with the null vector satisfy the linear differential equation known as the BPZ equation prefixed with Belavin, Polyakov and Zamolodchikov [7]. As the replica partition function $Z_{A}\left(q\right)$ is given by the correlation function of the twist operators, this correlation function can be obtained by solving the BPZ equation. Further, we show that the solution is consistent with the structure of the sub-system. For the two disjoint intervals, the BPZ equation is equivalent to the Heun’s differential equation. We determine the ERE by imposing an appropriate condition on the monodromy matrices of the Heun’s differential equation, and find the two saddle points that were obtained by the previous studies [4,5,6]. However, we will point out that these two saddle points should be treated carefully when applying the q→1 limit for the large *c*, because they contribute comparably, but not necessarily equally to $Z_{A}\left(q\right)$, which can be understood by considering the quantum state corresponding to multiple saddle points. The ERE is obtained by the Born rule.

This paper is structured as follows. In Section 2, we will review the replica trick and establish the relationship between the geometry of the replica manifold and the correlation function related to the ERE. In Section 3, we discuss how to treat multiple saddle points. In Section 4, we show the EREs for q∼1 based on the conditions specified in Section 3. Finally, Section 5 is the conclusion.

## 2. Entanglement Renyi Entropy (ERE) and Replica Trick

We will review the replica trick and the ERE of two disjoint intervals $A=\left[z_{1},z_{2}\right]\cup \left[z_{3},z_{4}\right]$ for a 2-dimensional CFT on the extended complex plane Σ=C∪{∞}[2,8]. To evaluate ${\mathrm{tr}}_{A}\left(\rho_{A}^{\;q}\right)$, it is useful to consider the replica manifold and the replica field theory. Figure 1 shows a schematic picture of the replica manifold $\Sigma_{A}\left(q\right)$ of the ERE for two disjoint intervals, the original manifold Σ with the twist operators $T_{q},{\tilde{T}}_{q}$, and the conformal map $w:\Sigma_{A}\left(q\right)\to \Sigma$. The left panel depicts the replica manifold $\Sigma_{A}\left(q\right)$ which comprises *q* sheets and a single field. The right panel depicts the replica field theory defined on Σ, which comprises *q* fields on the single sheet with the twist operators. The replica field theory provides the equivalent partition function to that of the theory on the replica manifold.

Let *Z* be the partition function of the CFT on Σ, and $Z_{A}\left(q\right)$ be partition function of the same CFT on the replica manifold $\Sigma_{A}\left(q\right)$. Because $Z_{A}\left(q\right)$ is constituted to satisfy ${\mathrm{tr}}_{A}\left(\rho_{A}^{\;q}\right)=Z_{A}\left(q\right)/Z^{q}$, the ERE $S_{A}\left(q\right)$ is calculated using the partition functions as follows:(3)$$\begin{array}{c} S_{A}\left(q\right)=\frac{1}{1-q}\left(\log Z_{A}\left(q\right)-q\log Z\right). \end{array}$$
The replica field theory is constructed so that the above ERE is expressed as the following 4-point correlation function on Σ:(4)$$\begin{array}{c} S_{A}\left(q\right)=\frac{1}{1-q}\log{\langle T_{q}\left(z_{1},{\bar{z}}_{1}\right){\tilde{T}}_{q}\left(z_{2},{\bar{z}}_{2}\right)T_{q}\left(z_{3},{\bar{z}}_{3}\right){\tilde{T}}_{q}\left(z_{4},{\bar{z}}_{4}\right)\rangle }_{\Sigma }, \end{array}$$
where $T_{q}$ and ${\tilde{T}}_{q}$ represent the primary twist operators with the same conformal weight $h_{q}=c\left(q^{2}-1\right)/\left(24q\right)$ and ${\bar{h}}_{q}=c\left(q^{2}-1\right)/\left(24q\right)$.

For a 2-dimensional CFT, there are some preferable properties to determine the correlation function. On the original manifold Σ, a correlation function incorporating the energy momentum tensor T(z), and the holomorphic part of the primary operators $O_{i}\left(z_{i}\right)$ with the conformal weight $h_{i}$ satisfies the following relation:(5)$$\begin{array}{c} {\langle T\left(z\right)\prod_{i=1}^{N}O_{i}\left(z_{i}\right)\rangle }_{\Sigma }=\sum_{i=1}^{N}\left[\frac{h_{i}}{{\left(z-z_{i}\right)}^{2}}+\frac{\partial_{z_{i}}}{z-z_{i}}\right]{\langle \prod_{i=1}^{N}O_{i}\left(z_{i}\right)\rangle }_{\Sigma }. \end{array}$$
In what follows, we abbreviate the anti-holomorphic part of operators, because we can obtain the equations for it immediately from those for the holomorphic part by adding the bar appropriately. For an arbitrary operator O(z), the following relation holds from the definition of the twist operators.
(6)$$\begin{array}{c} \frac{{\langle O\left(z\right)T_{q}\left(z_{1}\right){\tilde{T}}_{q}\left(z_{2}\right)T_{q}\left(z_{3}\right){\tilde{T}}_{q}\left(z_{4}\right)\rangle }_{\Sigma }}{{\langle T_{q}\left(z_{1}\right){\tilde{T}}_{q}\left(z_{2}\right)T_{q}\left(z_{3}\right){\tilde{T}}_{q}\left(z_{4}\right)\rangle }_{\Sigma }}=q{\langle O\left(z\right)\rangle }_{{\tilde{\Sigma }}_{A}\left(q\right)}, \end{array}$$
where ${\tilde{\Sigma }}_{A}\left(q\right)$ is one of the *q* sheets of the replica manifold $\Sigma_{A}\left(q\right)$ and we use the same complex coordinate on both ${\tilde{\Sigma }}_{A}\left(q\right)$ and Σ. If we obtain the conformal transformation $w\left(z\right):\;\Sigma_{A}\left(q\right)\to \Sigma$, the energy momentum tensor on the replica manifold is given by the Schwarzian derivative of w(z) as follows:(7)$$\begin{array}{c} {\langle T\left(z\right)\rangle }_{{\tilde{\Sigma }}_{A}\left(q\right)}=\frac{c}{12}\left[\frac{w^{\prime \prime \prime }\left(z\right)}{w^{\prime \prime }\left(z\right)}-\frac{3}{2}{\left(\frac{w^{\prime \prime }\left(z\right)}{w^{\prime }\left(z\right)}\right)}^{2}\right]. \end{array}$$
From Equations (Equation 5)–(Equation 7), we obtain the following differential equation which relates the conformal transformation, the conformal weight, and the 4-point correlation function as follows:(8)$$\begin{array}{c} \frac{qc}{12}\left[\frac{w^{\prime \prime \prime }\left(z\right)}{w^{\prime \prime }\left(z\right)}-\frac{3}{2}{\left(\frac{w^{\prime \prime }\left(z\right)}{w^{\prime }\left(z\right)}\right)}^{2}\right]=\sum_{i=1}^{4}\left(\frac{h_{q}}{{\left(z-z_{i}\right)}^{2}}-\frac{c_{i}}{z-z_{i}}\right), \end{array}$$
where $c_{i}=-\partial_{z_{i}}\log{\langle T_{q}\left(z_{1}\right){\tilde{T}}_{q}\left(z_{2}\right)T_{q}\left(z_{3}\right){\tilde{T}}_{q}\left(z_{4}\right)\rangle }_{\Sigma }$. The global conformal symmetry restricts the correlation function as
(9)$$\begin{array}{c} \;{\langle T_{q}\left(z_{1}\right){\tilde{T}}_{q}\left(z_{2}\right)T_{q}\left(z_{3}\right){\tilde{T}}_{q}\left(z_{4}\right)\rangle }_{\Sigma }={\left(z_{3}-z_{1}\right)}^{-2h_{q}}{\left(z_{4}-z_{2}\right)}^{-2h_{q}}{\langle T_{q}\left(0\right){\tilde{T}}_{q}\left(x\right)T_{q}\left(1\right){\tilde{T}}_{q}\left(\infty \right)\rangle }_{\Sigma } \end{array}$$
(10)$$\begin{array}{c} \sum_{i}c_{i}=0,\;\;\;\sum_{i}c_{i}z_{i}=4h_{q},\;\;\;\sum_{i}c_{i}z_{i}^{2}=2h_{q}\sum_{i}z_{i}, \end{array}$$
where $x=\left(z_{4}-z_{3}\right)\left(z_{2}-z_{1}\right){\left(z_{4}-z_{2}\right)}^{-1}{\left(z_{3}-z_{1}\right)}^{-1}$ denotes one of the cross ratios. Therefore, it is enough to deal with the following equation:(11)$$\begin{array}{cc} \; & \frac{w^{\prime \prime \prime }\left(z\right)}{w^{\prime \prime }\left(z\right)}-\frac{3}{2}{\left(\frac{w^{\prime \prime }\left(z\right)}{w^{\prime }\left(z\right)}\right)}^{2}=\frac{12h_{q}}{qc}Q\left(q,z\right), \end{array}$$(12)$$\begin{array}{cc} \; & Q\left(q,z\right)=\frac{1}{z^{2}}+\frac{1}{{\left(z-x\right)}^{2}}+\frac{1}{{\left(z-1\right)}^{2}}+2\left(\frac{1}{z}-\frac{1}{z-1}\right)+\frac{2a(q,x)}{z(z-x)(z-1)}, \end{array}$$
where a(q,x) is defined as:(13)$$\begin{array}{c} \frac{2h_{q}\;a\left(q,x\right)}{x(1-x)}=-\partial_{x}\log\langle T_{q}\left(0\right){\tilde{T}}_{q}\left(x\right)T_{q}\left(1\right){\tilde{T}}_{q}\left(\infty \right)\rangle . \end{array}$$
The solution of the third order non-linear differential Equation (Equation 11) for w(z) is described as:(14)$$\begin{array}{c} w\left(z\right)=\frac{\alpha \Psi_{1}\left(z\right)+\beta \Psi_{2}\left(z\right)}{\gamma \Psi_{1}\left(z\right)+\delta \Psi_{2}\left(z\right)},\;\;\;\alpha \delta -\beta \gamma =1, \end{array}$$
where $\Psi_{1}\left(z\right),\Psi_{2}\left(z\right)$ are the linearly independent solutions of the following linear differential equation:(15)$$\begin{array}{c} \frac{d^{2}}{dz^{2}}\Psi \left(z\right)+\frac{6h_{q}}{qc}Q\left(q,z\right)\Psi \left(z\right)=0. \end{array}$$
We can confirm that Equation (Equation 14) is the solution of Equation (Equation 11) by substituting it. Therefore, the ERE is equivalent to the correlation function of the twist operators, the energy momentum tensor on the replica manifold, the conformal map $w\left(z\right):\;\Sigma_{A}\left(q\right)\to \Sigma$, and the linearly independent solutions of Equation (Equation 15). Thus, the ERE can be obtained by evaluating any one of these entities. Next, we will evaluate a(q,x) for the semi-classical Liouville CFT. The function a(q,x) is comparable to the derivative of the ERE. In what follows, we call a(q,x) as the derivative of the ERE. To evaluate the derivative of the ERE a(q,x) for the semi-classical Liouville CFT, we solve Equation (Equation 15) with the condition that Ψ(z) goes to the next or previous sheet when crossing the sub-region *A*, as depicted in the left panel of Figure 1. Considering the twist operators, this condition implies that Ψ(z) is a *q* valued function on Σ and the phase of Ψ(z) varies with ±2π/q when Ψ(z) goes around the twist operators $T_{q}$ and ${\tilde{T}}_{q}$.

As a practice of the above procedure, let us consider the ERE of the single interval A=[u,v]. In this system, two twist operators are inserted at $z_{1}=u$ and $z_{2}=v$. From the global conformal symmetry, we can immediately obtain ${\langle T_{q}\left(u\right){\tilde{T}}_{q}\left(v\right)\rangle }_{\Sigma }\propto {\left(u-v\right)}^{-2h_{q}}$ without any other conditions. Subsequently, we obtain the derivative of the ERE $c_{1}=-c_{2}=2h_{q}\;{\left(u-v\right)}^{-1}$ from the definition $c_{i}=-\partial_{z_{i}}\log{\langle T_{q}\left(u\right){\tilde{T}}_{q}\left(v\right)\rangle }_{\Sigma }$ and the linearized equation corresponding to Equation (Equation 15) as:(16)$$\begin{array}{cc} \; & \frac{d^{2}}{dz^{2}}\Psi \left(z\right)+\frac{6h_{q}}{qc}Q\left(q,z\right)\Psi \left(z\right)=0, \end{array}$$(17)$$\begin{array}{cc} \; & Q\left(q,z\right)=\frac{1}{{\left(z-u\right)}^{2}}+\frac{1}{{\left(z-v\right)}^{2}}-\frac{2{\left(u-v\right)}^{-1}}{z-u}+\frac{2{\left(u-v\right)}^{-1}}{z-v}. \end{array}$$
The solution for this equation and the corresponding conformal map are determined as follows:(18)$$\begin{array}{cc} \; & \Psi \left(z\right)={\left(z-u\right)}^{\frac{1}{2}\left(1\pm \sqrt{1-\frac{24h_{q}}{qc}}\right)}{\left(z-v\right)}^{\frac{1}{2}\left(1\mp \sqrt{1-\frac{24h_{q}}{qc}}\right)}, \end{array}$$(19)$$\begin{array}{cc} \; & w\left(z\right)=\frac{\alpha {\left(z-u\right)}^{\sqrt{1-\frac{24h_{q}}{qc}}}+\beta {\left(z-v\right)}^{\sqrt{1-\frac{24h_{q}}{qc}}}}{\gamma {\left(z-u\right)}^{\sqrt{1-\frac{24h_{q}}{qc}}}+\delta {\left(z-v\right)}^{\sqrt{1-\frac{24h_{q}}{qc}}}}. \end{array}$$
Because Ψ(z) and w(z) are *q* valued functions on Σ, the local behavior of the conformal map should be consistent with $w\left(z∼u\right)∼{\left(z-u\right)}^{\pm 1/q}$; we further find the conformal weight $h_{q}=c\left(q^{2}-1\right)/\left(24q\right)$ again from the condition $\pm 1/q=\sqrt{1-24h_{q}/\left(qc\right)}$. Note that if α=δ=1,β=γ=0, we retrieve the well known conformal map $w\left(z\right)={\left(z-u\right)}^{1/q}{\left(z-v\right)}^{-1/q}$. This method works extraordinarily in this example because the behavior of the solutions Ψ(z) is completely determined by the conformal weight of the twist operators owing to the global conformal symmetry. However, because general multi-point correlation functions depend on the characteristics of each CFT, we can at most determine the local behavior of Ψ(z) without applying any other conditions related to the global structure of Σ. Therefore, an additional condition is required to be imposed to determine the global behavior of Ψ(z). We evaluate the ERE for the large *c* Liouville CFT on the condition that the Ψ(z) in Equation (Equation 15) serves as the 1-point correlation function on the replica manifold.

## 3. ERE with Multiple Saddle Points

In this section, we discuss the treatment of the ERE in the semi-classical approximation, within, the saddle points of the partition function represent the path integral in Equation (Equation 3). According to previous studies [4,5,6], the derivative of the ERE is given by a(q∼1,x)=1−x,−x from Equation (Equation 15) for the large *c* Liouville CFT. We often come across the statement [9] that the leading term of the derivative of the ERE for the large *c* limit is proportional to (q−1)c, then (q−1)c must be large enough for the saddle point approximation, and only the minimal-valued action contributes to the path integral for the partition function. However, we show that all the saddle points may comparably contribute to the ERE for q∼1. At least, we point out that only one of them does not represent the EE. First, we consider the case of two saddle points for two disjoint intervals system. From the saddle point approximation, the partition function $Z_{A}\left(q\right)$ is described as follows:(20)$$\begin{array}{c} Z_{A}\left(q\right)=p_{1}\exp\left[-I_{A,1}\left(q,x,\bar{x}\right)\right]+p_{2}\exp\left[-I_{A,2}\left(q,x,\bar{x}\right)\right], \end{array}$$
where $\bar{x}=\left({\bar{z}}_{4}-{\bar{z}}_{3}\right)\left({\bar{z}}_{2}-{\bar{z}}_{1}\right){\left({\bar{z}}_{4}-{\bar{z}}_{2}\right)}^{-1}{\left({\bar{z}}_{3}-{\bar{z}}_{1}\right)}^{-1}$ denotes one of the cross ratios, $p_{1},p_{2}$ are some constants and $I_{A,1}\left(q,x,\bar{x}\right),\;I_{A,2}\left(q,x,\bar{x}\right)$ are classical actions. Even after taking the large *c* limit, the normalization condition $\lim_{q\to 1}{\mathrm{tr}}_{A}\left(\rho_{A}^{\;q}\right)=1$ must hold. This means $\lim_{q\to 1}\left(Z_{A}\left(q\right)/Z^{q}\right)=1$, and then
(21)$$\begin{array}{c} \lim_{q\to 1}\left(\;p_{1}\exp\left[qI-I_{A,1}\left(q,x,\bar{x}\right)\right]+p_{2}\exp\left[qI-I_{A,2}\left(q,x,\bar{x}\right)\right]\;\right)=1, \end{array}$$
where we assume that I=−logZ is the unique Euclidean classical action of the original theory; it is the *c* order term. Because the replica field theory involves the *q* replicated field of the original theory, the effective action $I_{A,i}\left(q,x,\bar{x}\right)$ may be comparable to $I_{A,i}\left(q,x,\bar{x}\right)=qI+O\left(q-1\right)$ for q∼1. Thus, $p_{1}+p_{2}=1$ from Equation (Equation 21). One may concern that the two saddle points merge into the saddle point of the original theory for q∼1 and become indistinguishable. As long as q≠1, even if *q* is infinitesimally close to 1, there exist the two topologically distinguishable configurations of the classical field Ψ(z) as observed in previous studies [4,5,6]. Thus, from the viewpoint of the path integral, there exist the two distinct saddle points, out of which the leading term behaves like the *c* order term, as long as q≠1. Then, each classical action are described as $I_{A,i}\left(q,x,\bar{x}\right)=qI+b_{i}\left(q,x\right)+{\bar{b}}_{i}\left(q,\bar{x}\right)$, $\lim_{q\to 1}b_{i}\left(q,x\right)=0$ and $\lim_{q\to 1}{\bar{b}}_{i}\left(q,\bar{x}\right)=0$. We will explain why the q−1 order term of the classical action is decomposed into the holomorphic and the anti-holomorphic part after deriving the ERE. Therefore, Equation (Equation 3) for an arbitrary *q* in the semi-classical limit becomes
(22)$$\begin{array}{c} S_{A}\left(q\right)=\frac{1}{1-q}\left(\log\sum_{i=1}^{2}p_{i}\exp\left[-b_{i}\left(q,x\right)-{\bar{b}}_{i}\left(q,\bar{x}\right)\right]\right). \end{array}$$
As a result, the term in the parenthesis is proportional to (q−1)c. However, note that the leading terms of the saddle points for $Z_{A}\left(q\right)$ are proportional to qI, and then Equation (Equation 22) is derived from the cancellation between qI, which originated from $\log Z_{A}\left(q\right)$ and one from qlogZ. Thus, the saddle point approximation is valid for a large *c* independent of the magnitude of (q−1)c. Equation (Equation 22) is consistent with the decomposition of the 4-point function into the conformal blocks. Thus, we can assume that the q−1 order term of the classical action is decomposed into the holomorphic and the anti-holomorphic part.

We obtained Equation (Equation 22) as the ERE in the semi-classical limit with an arbitrary *q* based on the assumption that the replica field theory has two saddle points for the large *c*. If we adopt a large enough, but finite *c* for the saddle point approximation and keep q−1 finite, we can consider that only the minimal saddle point contributes to the ERE for a large (q−1)c. Conversely, if (q−1)c∼0 with large finite *c* and q∼1, the ERE becomes
(23)$$\begin{array}{c} \begin{array}{cc} S_{A}\left(q∼1\right)\; & ∼\frac{1}{1-q}\log\sum_{i=1}^{2}p_{i}\left(1-b_{i}\left(q,x\right)-{\bar{b}}_{i}\left(q,\bar{x}\right)\right)\\ \; & ∼\frac{1}{1-q}\left[\log\sum_{i=1}^{2}p_{i}-\sum_{i=1}^{2}\frac{p_{i}\left(b_{i}\left(q,x\right)+{\bar{b}}_{i}\left(q,\bar{x}\right)\right)}{\sum_{i=1}^{2}p_{i}}\right]. \end{array} \end{array}$$
Owing to the normalization condition of $p_{1}+p_{2}=1$, the EE is defined as:(24)$$\begin{array}{c} S_{A}=\lim_{q\to 1}S_{A}\left(q\right)=\lim_{q\to 1}\sum_{i=1}^{2}\frac{p_{i}\;\left(b_{i}\left(q,x\right)+{\bar{b}}_{i}\left(q,\bar{x}\right)\right)}{q-1}. \end{array}$$
Thus, the EE determined in the semi-classical limit is a summation of all the (q−1)c order terms of the classical actions. The following two nuances should be noted: First, in the semi-classical approximation, the leading terms of the classical action of the two partition functions cancel each other owing to the structure of the replica theory and the normalization condition of the density matrix. Second, the q→1 limit is adopted so that (q−1)c∼0 is satisfied. Therefore, because of the exquisite relationship between the two limits, the multiple saddle points comparably contribute to the EE with the contribution weights $p_{i}$. The above cautions are specific to the EE in the semi-classical approximation. Thus, we do not need to worry about it in other scenarios, such as the thermal phase transition of physical systems.

We identify the relation between the derivative of the ERE a(q,x) and the order (q−1)c term of the classical action $b_{i}\left(q,x\right)$, and then determine $p_{1}$ and $p_{2}$. We should pay attention for the quantum state of the replica field theory to relate them. As there are two classical saddle points, it is natural that the replica field theory also has the two quantum states corresponding to them. We assume that the quantum state of the replica field theory is expressed as $|\Omega \rangle =\sqrt{p_{1}}|\Omega_{1}\rangle +\sqrt{p_{2}}|\Omega_{2}\rangle$, $\langle \Omega_{1}|\Omega_{2}\rangle ∼0$, where $|\Omega_{1}\rangle$ and $|\Omega_{2}\rangle$ represent the states corresponding to the respective classical actions in the semi-classical limit. Subsequently, the partition function is described as $Z_{A}\left(q\right)=p_{1}Z_{A,1}\left(q\right)+p_{2}Z_{A,2}\left(q\right)$, where we defined $Z_{A}\left(q\right)=\langle \Omega |\Omega \rangle ,\;Z_{A,i}\left(q\right)=\langle \Omega_{i}|\Omega_{i}\rangle$. We can associate the weights $p_{1}$ and $p_{2}$ to the probability amplitude of $|\Omega_{1}\rangle$ and $|\Omega_{2}\rangle$, respectively. According to the above argument, Equation (Equation 13) is written as:(25)$$\begin{array}{c} \frac{2h_{q}}{x(1-x)}a\left(q,x\right)=-\frac{p_{1}\partial_{x}Z_{A,1}\left(q\right)+p_{2}\partial_{x}Z_{A,2}\left(q\right)}{Z_{A}\left(q\right)}=-\frac{p_{1}\partial_{x}e^{-b_{1}\left(q,x\right)}+p_{2}\partial_{x}e^{-b_{2}\left(q,x\right)}}{p_{1}e^{-b_{1}\left(q,x\right)}+p_{2}e^{-b_{2}\left(q,x\right)}}. \end{array}$$
Because the ERE for q∼1 is equivalent to the summation of the derivative of the classical actions, as described in Equation (Equation 24), it is natural that the derivative of the ERE also decomposes into $a\left(q,x\right)=p_{1}a_{1}\left(q,x\right)+p_{2}a_{2}\left(q,x\right)$ at least for q∼1. In particular, we relate $a_{i}\left(q,x\right)$ and $b_{i}\left(q,x\right)$ as follows:(26)$$\begin{array}{c} \frac{2h_{q}}{x(1-x)}a_{i}\left(q,x\right)=-\frac{\partial_{x}e^{-b_{i}\left(q,x\right)}}{p_{1}e^{-b_{1}\left(q,x\right)}+p_{2}e^{-b_{2}\left(q,x\right)}}∼\partial_{x}b_{i}\left(q∼1,x\right) \end{array}$$
In the same way, we also relate ${\bar{a}}_{i}\left(q,\bar{x}\right)$ and ${\bar{b}}_{i}\left(q,\bar{x}\right)$. Thus, we find $2h_{q}{\bar{x}}^{-1}{\left(1-\bar{x}\right)}^{-1}{\bar{a}}_{i}\left(q,\bar{x}\right)∼-\partial_{\bar{x}}\log Z_{A,i}\left(q\right)$, and we can regard this as the definition of ${\bar{a}}_{i}\left(q,\bar{x}\right)$ for q∼1. On the above identification, the ERE is described as:(27)$$\begin{array}{c} S_{A}\left(q\right)∼\frac{2h_{q}}{q-1}\sum_{i=1}^{2}p_{i}\;\left(\int \frac{a_{i}\left(q,x\right)}{x(1-x)}dx+\int \frac{{\bar{a}}_{i}\left(q,\bar{x}\right)}{\bar{x}\left(1-\bar{x}\right)}d\bar{x}\right) \end{array}$$
Note that the two candidates of the derivative of the ERE a(q∼1,x)=1−x,−x are obtained just by analyzing Equation (Equation 15) independent of the quantum state. Therefore, we assume
(28)$$\begin{array}{c} p_{1}a_{1}\left(q∼1,x\right)=1-x,\;\;\;p_{2}a_{2}\left(q∼1,x\right)=-x. \end{array}$$
Next, we determine the weights $p_{1}$ and $p_{2}$ because we can only obtain $p_{i}a_{i}\left(q,x\right)$ and not $a_{i}\left(q,x\right)$ itself. Furthermore, the ERE is described by the 4-point function of the twist operators Equation (Equation 4). Consider the 4-point correlation function $G_{1234}\left(x\right)={\langle \phi_{1}\left(0\right)\phi_{2}\left(x\right)\phi_{3}\left(1\right)\phi_{4}\left(\infty \right)\rangle }_{\Sigma }$, where $\phi_{i}$ is a general operator. Because $G_{1234}\left(x\right)$ is independent of the way of the operator product expansion, and $G_{1234}\left(x\right)$ exhibits the crossing symmetry $G_{1234}\left(x\right)=G_{3214}\left(1-x\right)$. The first and the third operators, in the ERE in Equation (Equation 4), are identical twist operators; therefore, we obtain $G_{1234}\left(x\right)=G_{3214}\left(x\right)$ in addition to $G_{1234}\left(x\right)=G_{1234}\left(1-x\right)$. Therefore, the ERE $S_{A}\left(q\right)$ in Equation (Equation 22) is invariant with the replacement x→1−x; thereby, allowing $p_{1}=p_{2}=1/2$ to be true. In this system, we can confirm that $x=\bar{x}$ and $a\left(q,x\right)=\bar{a}\left(q,\bar{x}\right)$. Finally, the EE of the two disjoint intervals for the large *c* Liouville CFT from Equation (Equation 24) is
(29)$$\begin{array}{c} S_{A}=\lim_{q\to 1}\frac{4h_{q}}{q-1}\int \frac{1-2x}{x(1-x)}dx=\frac{c}{3}\log\frac{x(1-x)}{\epsilon^{2}}, \end{array}$$
where ϵ denotes the UV cut off scale. Consequently, the obtained EE is equivalent to that of the free compactified boson at the leading order of the large *c*. Thus, it shall not be in contradiction to any postulate of the CFT. Note that we do not need the weights $p_{i}$ to calculate the EE, and we exploited the symmetry between the two saddle points to determine the weights $p_{i}$. In general, we need some extra information to evaluate the weights $p_{i}$ and the ERE. If we obtain a complex valued saddle point and its complex conjugated one, we can assign $p_{1}=p_{2}=1/2$ for the ERE to be real valued [10]. Moreover, for non-static systems, $p_{i}$ may be time-dependent. It is possible that the contribution of the dominant saddle point varies with time due to the time dependence of $p_{i}$. The weights $p_{i}$ are the coefficients of the conformal block expansion of the multi-point function; therefore, they may be evaluated with AGT correspondence and related techniques [11,12].

## 4. Determination of ERE for the Semi-Classical Liouville CFT

In this section, we see that Ψ(z) in Equation (Equation 15) should behave as the 1-point correlation function on the replica manifold for the large *c* Liouville CFT, and then determine the ERE of the two disjoint intervals. The Liouville CFT contains the degenerate operator, and then the corresponding BPZ equation helps us to analyze the structure of the correlation functions [3]. Let $\psi_{\chi }\left(z\right)$ denote the light degenerate operator corresponding to the level 2 light null vector with the conformal weight $h_{\chi }$; wherein, the BPZ equation holds:(30)$$\begin{array}{c} \left[\frac{3\partial_{z}^{2}}{2(2h_{\chi }+1)}-\sum_{i=1}^{4}\left(\frac{h_{q}}{{\left(z-z_{i}\right)}^{2}}+\frac{\partial_{z_{i}}}{z-z_{i}}\right)\right]\langle \psi_{\chi }\left(z\right)T_{q}\left(z_{1}\right){\tilde{T}}_{q}\left(z_{2}\right)T_{q}\left(z_{3}\right){\tilde{T}}_{q}\left(z_{4}\right)\rangle =0 \end{array}$$
As we treat the *q*-replicated Liouville CFT, the central charge is *q* times that of the original theory, that is, $h_{\chi }=\left(5-qc+\sqrt{\left(qc-1)(qc-25\right)}\right)/16$. We can choose $\left(z_{1},z_{2},z_{3},z_{4}\right)=\left(0,x,1,\infty \right)$ without the loss of generality. In the large *c* semi-classical limit, we can rewrite this equation in a simple form through the following steps. The conformal weight $h_{\chi }$ is $h_{\chi }=-1/2-9/\left(2qc\right)+O\left(c^{-2}\right)$ and $\psi_{\chi }\left(z\right)$ is a light operator whose expectation value can be considered as a 1-point correlation function $\Psi_{\chi }\left(z\right)$ on the replica manifold. This means that the above 5-point correlation function behaves as follows:(31)$$\begin{array}{cc} \; & \Psi_{\chi }\left(z\right)=\frac{{\langle \psi_{\chi }\left(z\right)T_{q}\left(0\right){\tilde{T}}_{q}\left(x\right)T_{q}\left(1\right){\tilde{T}}_{q}\left(\infty \right)\rangle }_{\Sigma }}{{\langle T_{q}\left(0\right){\tilde{T}}_{q}\left(x\right)T_{q}\left(1\right){\tilde{T}}_{q}\left(\infty \right)\rangle }_{\Sigma }} \end{array}$$(32)$$\begin{array}{cc} \; & ⟹{\langle \psi_{\chi }\left(z\right)T_{q}\left(0\right){\tilde{T}}_{q}\left(x\right)T_{q}\left(1\right){\tilde{T}}_{q}\left(\infty \right)\rangle }_{\Sigma }=\Psi_{\chi }\left(z\right)\;\sum_{i}p_{i}\;e^{-I_{A,i}\left(q,x\right)}. \end{array}$$
Thus, we will deal with the following equation assuming that $\Psi_{\chi }\left(z\right)$ behaves as the 1-point correlation function on the replica manifold: (33)$$\begin{array}{cc} \; & \frac{d^{2}}{dz^{2}}\Psi_{\chi }\left(z\right)+\frac{q^{2}-1}{4q^{2}}Q\left(q,z\right)\Psi_{\chi }\left(z\right)=0, \end{array}$$(34)$$\begin{array}{cc} \; & Q\left(q,z\right)=\frac{1}{z^{2}}+\frac{1}{{\left(z-x\right)}^{2}}+\frac{1}{{\left(z-1\right)}^{2}}+2\left(\frac{1}{z}-\frac{1}{z-1}\right)+\frac{2a(q,x)}{z(z-x)(z-1)}, \end{array}$$(35)$$\begin{array}{cc} \; & a\left(q,x\right)=-\frac{12q}{c(q^{2}-1)}x\left(1-x\right)\partial_{x}\log Z_{A}\left(q\right). \end{array}$$
$\Psi_{\chi }\left(z\right)$ satisfies the same differential equation as Equation (Equation 15), but now we have an additional global condition that $\Psi_{\chi }\left(z\right)$ behaves as a 1-point correlation function on the replica manifold. Furthermore, we evaluate a(q,x) for q∼0 first as we can find an analytical expression of $\Psi_{\chi }\left(z\right)$ using the WKB approximation, and then numerically evaluate a(q,x) for q∼1.

First, as just a practice, we calculate the ERE for q∼0 using the WKB method because it enables in for understanding the relation between the structure of the replica manifold and the global behavior of $\Psi_{\chi }\left(z\right)$ on it. Consider the following WKB solution of Equation (Equation 33) in the leading order of the WKB approximation for q∼0:(36)$$\begin{array}{c} \Psi_{\chi }\left(z\right)=\frac{1}{Q{\left(q,z\right)}^{1/4}}\exp\left[\pm \frac{1}{2q}\int^{z}\sqrt{Q(q,\zeta )}d\zeta \right]. \end{array}$$
As we have the integral expression for Ψ(z), it is easy to analyze its global behavior, which is determined by the residues of $\sqrt{Q(q,z)}$. Note that we can rewrite $\sqrt{Q(q,z)}$ as
(37)$$\begin{array}{c} \sqrt{Q(q,z)}=\frac{\sqrt{z^{4}-2z^{3}+2z^{2}-2xz+x^{2}+2a\left(q,x\right)z\left(z-x\right)\left(z-1\right)}}{z(z-x)(z-1)}. \end{array}$$
The residues of $\sqrt{Q(q,z)}$ at z=0,x,1 are ±1 independent of a(q,x). From the requirement that Ψ(z) behaves as a 1-point correlation function on the replica manifold as depicted in Figure 1, a(q,x) is determined so that $\sqrt{Q(q,z)}$ transforms into a rational function and its Riemann surface is single sheeted, that is, $\mathrm{Res}\sqrt{Q(q,z=0)}=-\mathrm{Res}\sqrt{Q(q,z=x)}=\mathrm{Res}\sqrt{Q(q,z=1)}$ should hold. Therefore, we find the unique derivative of the ERE a(q,x)=1−2x, and then, $\sqrt{Q(q,z)}$ and the ERE for q∼0 is determined as follows:(38)$$\begin{array}{c} \sqrt{Q(q∼0,z)}=\pm \left(\frac{1}{z}-\frac{1}{z-x}+\frac{1}{z-1}\right), \end{array}$$(39)$$\begin{array}{c} S_{A}\left(q∼0\right)=\lim_{q\to 0}\frac{4h_{q}}{q-1}\int \frac{1-2x}{x(1-x)}dx=\frac{c}{6q}\log\frac{x(1-x)}{\epsilon^{2}}. \end{array}$$
We can express the conformal map as $w\left(z\right)=z^{\frac{1}{q}}{\left(z-x\right)}^{-\frac{1}{q}}{\left(z-1\right)}^{\frac{1}{q}}$; we obtain the energy momentum tensor for q∼0 as follows:(40)$$\begin{array}{c} \begin{array}{cc} \frac{12}{q\;c}T\left(z\right)\; & =\frac{q^{2}-1}{2q^{2}z^{2}}+\frac{q^{2}-1}{2q^{2}{\left(z-x\right)}^{2}}+\frac{q^{2}-1}{2q^{2}{\left(z-1\right)}^{2}}-\frac{\left(q^{2}-1\right)\left(x+z-1\right)}{q^{2}z\left(z-x\right)\left(z-1\right)}-\frac{6(x-1)x}{{\left(z^{2}-2xz+x\right)}^{2}}\\ \; & ∼-\frac{1}{2q^{2}}\left[\frac{1}{z^{2}}+\frac{1}{{\left(z-x\right)}^{2}}+\frac{1}{{\left(z-1\right)}^{2}}+2\left(\frac{1}{z}-\frac{1}{z-1}\right)+\frac{2(1-2x)}{z(z-x)(z-1)}\right] \end{array} \end{array}$$

The form of this energy momentum tensor is consistent with Equation (Equation 5) for q∼0, that is, it has the same poles. For a finite *q*, the sub-leading terms of the WKB solution may cancel the extra poles at $z^{2}-2xz+x=0$. Additionally, for a(q,x)=±1, $\sqrt{Q(q,z)}$ also becomes a rational function:(41)$$\begin{array}{cc} \; & a\left(q,x\right)=1\;⟺\;\sqrt{Q(q∼0,z)}=\pm \left(\frac{1}{z}-\frac{1}{z-x}-\frac{1}{z-1}\right), \end{array}$$(42)$$\begin{array}{cc} \; & a\left(q,x\right)=-1\;⟺\;\sqrt{Q(q∼0,z)}=\pm \left(\frac{1}{z}+\frac{1}{z-x}-\frac{1}{z-1}\right). \end{array}$$
From the relative sign of the poles, the 4-point correlation functions corresponding to them are given as:(43)$$\begin{array}{cc} \; & a\left(q,x\right)=1\;⟺\;{\langle T_{q}\left(z_{1}\right){\tilde{T}}_{q}\left(z_{2}\right){\tilde{T}}_{q}\left(z_{3}\right)T_{q}\left(z_{4}\right)\rangle }_{\Sigma }, \end{array}$$(44)$$\begin{array}{cc} \; & a\left(q,x\right)=-1\;⟺\;{\langle T_{q}\left(z_{1}\right)T_{q}\left(z_{2}\right){\tilde{T}}_{q}\left(z_{3}\right){\tilde{T}}_{q}\left(z_{4}\right)\rangle }_{\Sigma }. \end{array}$$
This practice clearly demonstrates the relation between each 4-point correlation function and the geometry of each replica manifold. We may be able to precisely analyze by considering the higher order term of the WKB solution. Thus, we confirm the one-to-one correspondence between each saddle point $a_{i}\left(q,x\right)$ and each replica manifold. However, we obtain multiple saddle points for the general *q*.

Second, we consider the q∼1 case. Let $\Phi \left(z\right)=g\left(x\right)\Psi_{\chi }\left(z\right)$ and $g\left(z\right)=z^{\frac{q-1}{2q}}{\left(z-x\right)}^{\frac{q+1}{2q}}{\left(z-1\right)}^{\frac{q-1}{2q}}$, then Equation (Equation 33) is transformed into the Heun’s differential equation as follows:(45)$$\begin{array}{cc} \; & \frac{d^{2}}{dz^{2}}\Phi \left(z\right)+\left(\frac{\gamma }{z}+\frac{\epsilon }{z-x}+\frac{\delta }{z-1}\right)\frac{d}{dz}\Phi \left(z\right)+\frac{\alpha \beta z-p}{z(z-x)(z-1)}\Phi \left(z\right)=0, \end{array}$$(46)$$\begin{array}{cc} \; & \alpha =1,\;\beta =1-\frac{1}{q},\;\gamma =1-\frac{1}{q},\;\delta =1-\frac{1}{q},\;\epsilon =1+\frac{1}{q}, \end{array}$$(47)$$\begin{array}{cc} \; & p=\frac{q-1}{2q^{2}}\left[1-2x+q-\left(q+1\right)a_{i}\left(q,x\right)\right] \end{array}$$
The solution of this equation is called as the Heun function. The Heun’s differential equation has four regular singular points at z=0,x,1,∞ and the Frobenius solutions of the Heun’s differential equation are known as the local Heun functions. For example, two independent local Heun functions around z=0 can be expressed as:(48)$$\begin{array}{cc} \; & \Phi (z∼0)∼\mathrm{HeunG}[x,p,\alpha ,\beta ,\gamma ,\delta ,z], \end{array}$$(49)$$\begin{array}{cc} \; & \Phi \left(z∼0\right)∼z^{1-\gamma }\mathrm{HeunG}\left[x,p+\left(1-\gamma \right)\left(\delta x+\epsilon \right),\alpha -\gamma +1,\beta -\gamma +1,2-\gamma ,\delta ,z\right], \end{array}$$
where the local Heun function is normalized as HeunG[x,p,α,β,γ,δ,z=0]=1 [13]. We denote a local Heun function near $z=z_{i}$ with the characteristic exponent *s* as $y_{z_{i}}^{s}\left(z\right)$. The connection matrix describes the relationships between the local Heun functions. For example, $y_{x}^{s}\left(z\right)$ and $y_{0}^{s}\left(z\right)$ are connected by the connection matrix $C_{x0}$ as follows:(50)$$\begin{array}{c} \left(\begin{array}{c} y_{x}^{0}\left(z\right)\\ y_{x}^{1-\epsilon }\left(z\right) \end{array}\right)=\frac{1}{W(y_{0}^{0},y_{0}^{1-\gamma })}\left(\begin{array}{cc} W(y_{x}^{0},y_{0}^{1-\gamma }) & W(y_{0}^{0},y_{x}^{0})\\ W(y_{x}^{1-\epsilon },y_{0}^{1-\gamma }) & W(y_{0}^{0},y_{x}^{1-\epsilon }) \end{array}\right)\left(\begin{array}{c} y_{0}^{0}\left(z\right)\\ y_{0}^{1-\gamma }\left(z\right) \end{array}\right), \end{array}$$
where $W\left(y_{x}^{0},y_{x}^{1-\epsilon }\right)=y_{x}^{0}\left(z\right)\partial_{z}y_{x}^{1-\epsilon }\left(z\right)-\partial_{z}y_{x}^{0}\left(z\right)y_{x}^{1-\epsilon }\left(z\right)$ is the Wronskian of $y_{x}^{0}\left(z\right)$ and $y_{x}^{1-\epsilon }\left(z\right)$ and the others are the same. The ratio of these Wronskians attains a constant value with respect to *z*, contrary to the Wronskians themselves. We utilized the Mathematica to calculate these Wronskians, see [14].

The derivative of the ERE a(q,x) determines the connection matrices. Additionally, we need to find the condition that the connection matrices must satisfy. To formulate it, consider the paths $P_{0x}$ and $P_{1x}$, which encircle the interval [0,x] or [x,1] once in the counterclockwise direction. Additionally, let $R_{0}=R_{x}^{-1}=R_{1}=R_{\infty }^{-1}=\mathrm{diag}\left(1,\exp\left[2\pi i/q\right]\right)$ and the connection matrix $C_{x0}$ be given by Equation (Equation 50) and the others be defined in the same manner. Then, the analytic continuation along $P_{0x}$ for the local Heun functions $y_{x}^{0}$ and $y_{x}^{1-\epsilon }$ are described as:(51)$$\begin{array}{c} \left(\begin{array}{c} y_{x}^{0}\left(z\right)\\ y_{x}^{1-\epsilon }\left(z\right) \end{array}\right)=M_{0x}\left(\begin{array}{c} y_{x}^{0}\left(z\right)\\ y_{x}^{1-\epsilon }\left(z\right) \end{array}\right), \end{array}$$
where we define the monodromy matrix $M_{0x}=C_{x0}R_{0}C_{0x}R_{x}$ as depicted in Figure 2.

Similarly, the analytic continuation along $P_{1x}$ is expressed comparable to the other monodromy matrix $M_{1x}=C_{x1}R_{1}C_{1x}R_{x}$. One may hope that both the monodromy matrices transform into the identity matrix like the WKB analysis for q∼0. However, both cannot transform into the identity matrix simultaneously for general *q* whatever a(q,x) is chosen. Instead, one of two should be the identity matrix, also known as the Schottky uniformization [4,15,16]. Moreover, it is trivial for the analytic continuation along the path which encircles all the four regular singular points once in the counterclockwise direction. Therefore, $M_{0x}=I$ is equivalent to $M_{\infty 1}=I$ because $C_{x1}M_{\infty 1}C_{1x}M_{0x}=I$ and $C_{1x}C_{x1}=I$. Comparably, $M_{1x}=I$ implies $M_{0\infty }=I$; thus, it is sufficient to deal with the monodromy matrices $M_{0x}$ and $M_{1x}$. On this condition, the monodromy matrices $M_{0x}$ and $M_{1x}$ are commutative. Thus, if we perform the analytical continuation via *q* times $P_{0x}$ and *q* times $P_{1x}$ for integer *q*, Φ(z) retains its original value because this is the first time that Φ(z) is back to the starting point from the viewpoint of the replica manifold. For 0<q∈Q, let q=t/u with $t,u\in Z^{+}$, while considering the analytical continuation via *u* times $P_{0x}$ and *u* times $P_{1x}$ in random order, the same discussion holds because Φ(z) is *t* times back to the starting point. Therefore, we accept the Schottky uniformization for an arbitrary q∈R, if the ERE is a continuous function with respect to *q*.

For an arbitrary *q* near x=0 or x=1, the derivative of the ERE behaves as a(q,x∼0)∼1 or a(q,x∼1)∼−1, respectively [5,6]. Then, we regard the former as $p_{1}a_{1}\left(q,x\right)$ if there are only two saddle points. We numerically calculate $p_{1}a_{1}\left(q,x\right)$ in case all the components of the commutation relation between the two monodromy matrices $\left[M_{0x},M_{1x}\right]$ vanish simultaneously. Then, we obtain the ERE from Equations (Equation 22) and (Equation 26) with $a_{1}\left(q,x\right)=-a_{2}\left(q,1-x\right)$.

Figure 3 shows $p_{1}a_{1}\left(q,x\right)$ and the ERE $S_{A}\left(q\right)$ for q∼1. For q→1, we can consider $p_{1}a_{1}\left(q\to 1,x\right)\to 1-x$. As mentioned before, we obtained the same EE as that of a compactified boson. Note that the central charge *c* should be large enough because Equation (Equation 22) is based on the saddle point approximation. For (q−1)c∼0, the EREs depend on *c* only linearly, and then the EREs with c=1 in Figure 3 is meaningful. It is difficult to compute the ERE not for q∼1. In particular, for q<0.5, the number of saddle points increases with a decreasing *q*, and we cannot determine the weights of the contribution for each saddle point to the ERE. Moreover, we cannot calculate each saddle point for small *q* owing to the lack of numerical accuracy. The WKB analysis could be considered to calculate the ERE for this region. The monodromy analysis using the AGT correspondence [17] or analytic expression of the connection matrices [18] may help for determining the ERE.

## 5. Conclusions

In this study, we reviewed the relationship between the ERE and the geometrical structure of the replica manifold and saw that some additional conditions must be imposed to determine the ERE of two disjoint intervals system in general. Then, we considered the treatment of the EE in the semi-classical approximation in general. Because of the exquisite relationship between the large *c* and q→1, we pointed out that the multiple saddle points contribute comparably to the EE. The leading terms of the classical action of the two partition functions $Z_{A}\left(q\right)$ and *Z* for large *c* cancel each other due to the structure of the replica theory and the normalization condition of the density matrix. For the case of general ERE, the method to evaluate the contribution weights of each saddle point is not known. Thus, we numerically evaluated the ERE of the two disjoint intervals for the large *c* Liouville CFT for q∼1 by analyzing the BPZ equation by satisfying the criterion that its solution behaves like a 1-point correlation function on a replica manifold. This condition is expressed by the condition that one of the monodromy matrices transforms into the identity matrix for any real number *q*.

In future work, it shall be of interest to reconsider ERE in other scenarios and entanglement measures. For instance, there is a growing interest in the reflected Renyi entropy, which signifies that the corresponding replica manifold exhibits a rather complex geometry [19]. Additionally, we can consider the ERE of a single interval on the torus as it is also well known for the expression of the Heun’s differential equation. Conversely, it would be interesting to evaluate the higher order terms of the WKB method and the large *c*. Considering the higher order terms of the WKB method for finite *q*, we may check the consistency between the WKB method and the numerical method for the ERE of the two disjoint intervals. For higher order corrections of large *c*, it may be necessary to evaluate the contribution of the cross term between multiple quantum states corresponding with each saddle point in case of multiple saddle points.

## Figures and Tables

**Figure 1 entropy-24-01758-f001:**
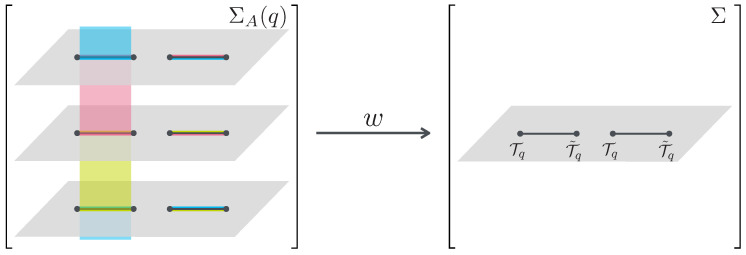
The left panel depicts the replica manifold $\Sigma_{A}\left(q\right)$ with q=3 sheets. Each sheet is connected to the same-colored lines. The right panel depicts the original manifold Σ with the twist operators $T_{q},{\tilde{T}}_{q}$ at the boundary of the sub-system. The conformal map *w* maps $\Sigma_{A}\left(q\right)$ to Σ.

**Figure 2 entropy-24-01758-f002:**
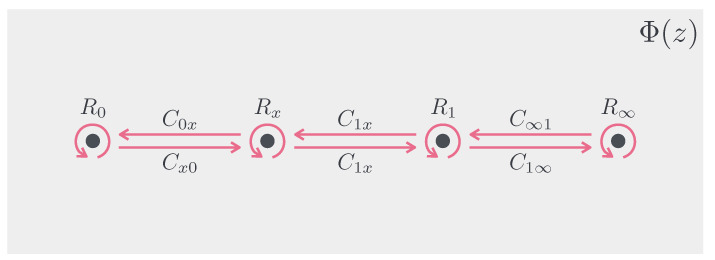
The dots represent z=0,x,1,∞ on Σ from left to right. The analytic continuation along each magenta line is described as a matrix, such as $C_{x0},\cdots$ and $R_{0},\cdots$. The paths $P_{0x}$ and $P_{1x}$ correspond to the monodromy matrices $M_{0x}=C_{x0}R_{0}C_{0x}R_{x}$ and $M_{1x}=C_{x1}R_{1}C_{1x}R_{x}$, respectively.

**Figure 3 entropy-24-01758-f003:**
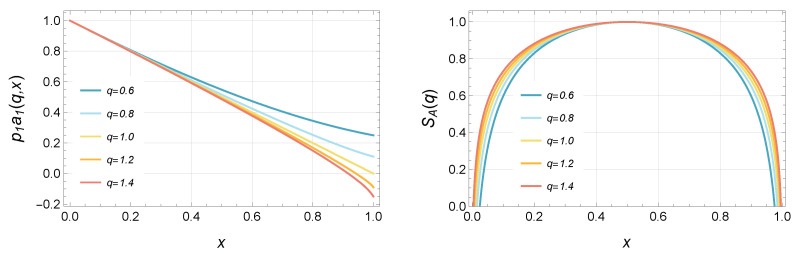
The left panel shows $p_{1}a_{1}\left(q,x\right)$ for q=0.6,0.8,1.0,1.2,1.4. The right panel shows the corresponding ERE normalized as $S_{A}\left(q\right)=1$ at x=1/2 with the central charge c=1 and the UV cutoff ϵ=0.1.

## Data Availability

Not applicable.

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
