# Peer review of "Entanglement Renyi Entropy of Two Disjoint Intervals for Large c Liouville Field Theory"

_entropy, 2022, doi:10.3390/e24121758_

Round 1
Reviewer 1 Report
Please see the report attached.

Reviewer 2 Report
Please see an attached file

Reviewer 3 Report
I am not convinced about the correctness of the main results obtained in this paper. In particular, the authors analyze an interesting question of understanding the contributions of subleading saddle points to the entanglement entropy in the limit they correctly describe is often difficult to analyze in the literature. However, their particular analysis suffers from various issues which have been pointed out previously in the literature. For example, Section 2 of the paper (https://arxiv.org/pdf/1407.2900.pdf) discusses an identical calculation to the one done in this paper, although in a holographic context. Importantly, the analysis leads to the wrong answer there and over time, it has been better understood in the literature that the correct answer arises from including replica symmetry breaking effects (e.g. https://arxiv.org/pdf/1911.11977.pdf). I believe the calculation done by the authors is incorrect for the same reason. Their Equation (20) is essentially analogous to Eq 2.6a in (https://arxiv.org/pdf/1407.2900.pdf).
Round 2
Reviewer 1 Report
The authors have now adequately answered my concerns.
However, the referee 3 has cited a very relevant reference (https://arxiv.org/abs/1407.2900) which must be cited in their manuscript appropriately. I am not entirely convinced by the author's responses to referee 3, but that shouldn't stop this manuscript from getting published.
Reviewer 2 Report
The authors have honestly revised the manuscript. I recommend it for publication.
Reviewer 3 Report
The authors have not addressed my concerns. They have pointed out that the references I provided are in a gravitational context as they indeed are. However, the analysis of entanglement entropy in a large c CFT is quite parallel to that in a gravitational theory. In particular, the results provided here should be applicable to a holographic theory. Since in those cases, we have evidence to show that the result is incorrect, it calls into question the analysis for non-holographic large c CFTs as well. Thus, I cannot accept the draft in its present form.
Round 3
Reviewer 1 Report
I am fine if the manuscript is now accepted for publication.
Reviewer 3 Report
The paper can be accepted in current form.